# Characteristics of postintubation dysphagia in ICU patients in the context of the COVID-19 outbreak: A report of 920 cases from a Brazilian reference center

Fernanda Chiarion Sassi[1☯], Ana Paula Ritto[1‡], Maíra Santilli de Lima[2‡], Cirley Novais Valente Junior[2‡], Paulo Francisco Guerreiro Cardoso[3‡], Bruno Zilberstein[4‡], Paulo Hilário Nascimento Saldiva[5‡], Claudia Regina Furquim de Andrade[1☯]*

1 Department of Physiotherapy, Speech-Language and Hearing Science and Occupational Therapy, School of Medicine, University of São Paulo, São Paulo, Brazil, 2 Division of Oral Myology, Hospital das Clínicas, School of Medicine, University of São Paulo, São Paulo, Brazil, 3 Department of Cardiopneumology, Thoracic Surgery Discipline, School of Medicine, University of São Paulo, São Paulo, Brazil, 4 Division of Digestive Surgery, Hospital das Clínicas, School of Medicine, University of São Paulo, São Paulo, Brazil, 5 Department of Pathology, School of Medicine, University of São Paulo, São Paulo, Brazil

☯ These authors contributed equally to this work.
‡ These authors also contributed equally to this work.
* clauan@usp.br

**Data Availability Statement:** The data that support the findings of this study are available from Comitê de Ética para Análise de Projetos de Pesquisa do

## Abstract

The purpose of this research was to identify risk factors that were independently related to the maintenance of a swallowing dysfunction in patients affected by critical COVID-19. We conducted a prospective observational cohort study of critical patients with COVID-19, who were admitted to a COVID-19 dedicated intensive care unit (ICU) and required prolonged orotracheal intubation (≥48 hours). Demographic and clinical data were collected at ICU admission and/or at hospital discharge or in-hospital death. Swallowing data was based on The Functional Oral Intake Scale (FOIS) and was collected at two distinct moments: initial swallowing assessment and at patient outcome. Patients were divided into two groups according to their FOIS level assigned on the last swallowing assessment: in-hospital resolved dysphagia–patients with FOIS levels 6 and 7; non-resolved dysphagia at hospital outcome–patients with FOIS levels 1 to 5. Nine hundred and twenty patients were included in our study. Results of the multivariate logistic regression model for the prediction of non-resolved dysphagia at hospital outcome in critical COVID-19 patients. indicated that increasing age (p = 0.002), severity at admission (p = 0.015), body mass index (p = 0.008), use of neuromuscular blockers (p = 0.028), presence of neurologic diseases (p = 0.038), presence of Diabetes Mellitus (p = 0.043) and lower FOIS levels on the initial swallowing assessment (p<0.001) were associated with higher chances of presenting dysphagia at hospital outcome. Critical patients with COVID-19 may experience post-acute COVID-19 dysphagia, indicating the need to prepare for the care/rehabilitation of these patients.

HCFMUSP but restrictions apply to the availability of these data, which were used under license for the current study, and so are not publicly available. Data are however available from the authors upon reasonable request and with permission of Comitê de Ética para Análise de Projetos de Pesquisa do HCFMUSP (cappesq.adm@hc.fm.usp.br). Our institution, Hospital das Clínicas da Faculdade de Medicina da Universidade de São Paulo, developed a data management plan concerning the general database that compiles information regarding demographic, clinical, laboratory, medical imaging and other biomarker data obtained during hospital admissions for COVID-19 and addressing the aspects related to the public use of the data. The plan outlines not only the data access policies for dissemination of scientific results by HCFMUSP researchers, but also the guiding criteria for the release of the data to open-access repositories, which will be done only after the publication and dissemination of the results of the initial analyses performed by our research groups and their direct collaborators. HCFMUSP will participate in the COVID Brazil Data-Sharing repository coordinated by The State of São Paulo Research Foundation (FAPESP), providing open access to hospital data related to COVID-19 hospitalizations.

**Funding:** The authors received no specific funding for this work.

## Introduction

The COVID-19 pandemic is currently one of the most serious global public health concerns. Until September 2021 234,157,747 people were infected globally as a result of the disease and 4,790,066 deaths were reported [1]. Although knowledge about the transmission, development and treatment of the disease has grown considerably since the outbreak of the pandemic in March 2020, a lot is yet unknown about the long-term consequences of COVID-19 [2]. There is, therefore, need to prepare for the care/rehabilitation of patients in the post-acute phase.

According to the literature, when looking at potential factors that could have long-term consequences on patients' health, intensive care unit (ICU) stay seems to be the most important [2]. A few studies have reported respiratory and heart dysfunction [3], severe muscle weakness and fatigue [4], psychological problems [5] and impaired functioning concerning mobility and activities of daily life [6] as main long-term consequences of COVID-19.

Dysphagia has also been identified as being associated to critical COVID-19 patients [7–13]. The negative effects of COVID-19 on swallowing seem to be associated to the critical form of the disease, need for prolonged intensive care, intubation or mechanical ventilation, tracheostomy or nasogastric tubes, and to patients with acute respiratory infection, pneumonia and respiratory insufficiency [8]. Studies have already identified that dysphagia tends to be highly prevalent in patients after prolonged intubation–i.e., up to 84% [14–16], about half of whom remain with swallowing difficulties on hospital discharge [14]. According to the literature, prolonged orotracheal intubation is typically defined as intubation for 48 hours or longer [17–24]. In addition, various central and peripheral neurologic complications related to COVID-19, such as polyneuropathy, myopathy, anosmia and ageusia, directly affect the swallowing network, making patients with COVID-19 more susceptible to developing swallowing disorders [8,25]. The importance of identifying and rehabilitating patients with swallowing disorders lies on the fact that dysphagia has been associated with negative patient outcome, including aspiration pneumonia, malnutrition, increased length of admission, increased readmission rate due to worsening of the case and higher mortality [26]. To the best of our knowledge, there is still lack of information regarding the comparison of swallowing abilities in critical COVID-19 and non-COVID-19 patients.

Considering that rehabilitation programs should be included in COVID-19 long-term care in order to minimize the level of potential disability, the main purpose of our research was to identify risk factors that were independently related to the maintenance of a swallowing dysfunction in patients affected by critical COVID-19 and who required prolonged mechanical ventilation after discharge from the ICU.

## Materials and methods

We conducted a prospective observational cohort study of critical patients with Corona virus disease (COVID-19) who were admitted to a COVID-19 dedicated ICU. The study protocol was approved by the Scientific and Ethic Committee of the Institution (*Comitê de Ética para Análise de Projetos de Pesquisa do HCFMUSP*), under the number 4.415.496. Data gathering procedures only began after obtaining written consent of all patients included in the study.

### Patient population

*Hospital das Clinicas* is a 2,200-bed public teaching hospital complex in Sao Paulo dedicated to treating high-complexity medical and surgical patients. In March 2020, its 900-bed central building started to operate as a special COVID-19 treatment center, receiving patients with moderate to severe symptoms. Additionally, its intensive care capacity was increased with the conversion of regular wards to ICUs, to a total of 300 dedicated ICU beds.

Patients were eligible for this study if they met all of the following criteria: (a) admission to the COVID-19 dedicated ICU from April 2020 to August 2021; (b) SARS-CoV-2 infection confirmed by either RT-PCR or serology (serology was used as a confirmatory test for probable COVID-19 cases not tested with RT-PCR within ten days after onset of symptoms); (c) submitted to prolonged orotracheal intubation ($\geq$48 hours) [17–24]; (d) bedside swallow evaluation (BSE) and swallow treatment requested by the primary treating physician and performed by a speech-language pathologist (SLP); (e) age $\geq$18 years; (f) clinical and respiratory stability; (g) score $\geq$13 points on the Glasgow Coma Scale; (h) patients in use of a tracheostomy tube had to remain with a deflated cuff during swallowing assessment.

Studies that have investigated the presence of post-extubation oropharyngeal dysphagia as early as 2–4 hours after extubation have pointed that a patient's risk for aspiration will decrease over the first 24 hours as a result of laryngeal recovery, improved respiratory function, and improved mental status [24]. Moreover, 10% of patients identified as presenting swallowing disorders 24 hours after extubation will present persistent dysphagia [16]. For all of these reasons, the protocol adopted for the present study determines that patients should be assessed 48 hours after extubation to ensure that those identified with a swallowing disorder presented real risk of pulmonary aspiration and, therefore, needed swallowing rehabilitation. Due to the protocols adopted during COVID-19 and the risk of contamination, gold standard identification of aspiration (i.e. videofluoroscopy or fiberoptic endoscopic evaluation of swallowing) was not possible. Assessing the swallowing of patients 48 hours after extubation excluded patients with transitory dysphagia.

## Measurements–clinical assessment of swallowing

The Functional Oral Intake Scale [27] is a 7-point ordinal scale developed to document the functional level of oral intake of food and liquid in patients with risk of dysphagia (Table 1). For this study, the patients' swallowing ability was used to assign a specific level on FOIS based on the clinical assessment of safety and/or efficiency of eating. The professional conducting this assessment had successfully passed specific training tests. The level on FOIS was determined based on the results obtained in the Dysphagia Risk Evaluation Protocol–DREP [15].

The DREP [15] is a bedside assessment protocol designed for the early detection of dysphagia risk. The protocol has already been validated in the specific literature demonstrating excellent validity with sensitivity at 92.9%, specificity at 75.0%, negative predictive values at 95.5% and an accuracy of 80.9% [28].

## Swallowing management

All patients underwent specific controlled swallowing and oral-motor rehabilitation sessions. Rehabilitation sessions were conducted based on current strategies of swallowing and oral-motor therapy. Patients were seen by a trained SLP for approximately 30 minutes per session.

**Table 1. FOIS items.**

| Level 1 | Nothing by mouth. |
|---|---|
| Level 2 | Tube dependent with minimal attempts of food or liquid. |
| Level 3 | Tube dependent with consistent oral intake of food or liquid. |
| Level 4 | Total oral diet of a single consistency. |
| Level 5 | Total oral diet with multiple consistencies, but requiring special preparation or compensations. |
| Level 6 | Total oral diet with multiple consistencies without special preparation, but with specific food limitations. |
| Level 7 | Total oral diet with no restrictions. |

Overall, treatment involved the use of direct and indirect therapy techniques to rehabilitate swallowing. Direct therapy involved controlled food offerings, even if in small volumes, for swallowing training, including the use of auxiliary tools such as straw, spoon or glass. Indirect therapy involved the use of exercises for oral motor training and sensomotor recovery with tongue and lip exercises. Compensatory treatment procedures such as postural changes (i.e., head rotation our upright positioning), adaptive measures including dietary consistency modifications, modifications in volume and tempo of food presentation, and specific swallowing techniques (e.g. supraglottic swallow) were also adopted whenever necessary in order to guarantee safe oral feeding and, consequently, the removal of the alternative feeding method.

## Severity of illness and clinical indicators

Severity of illness was obtained using the hospital database and included data obtained from the Simplified Acute Physiology Score (SAPS-3) [29], calculated at the time of admission to the ICU. Other clinical data included in the study to determine possible factors associated to the return to a safe swallowing function were: age; sex; mortality; body mass index on admission; comorbidities; D-dimer level on admission; PaO2/FiO2 ratio after orotracheal intubation; duration of medication—corticosteroids and neuromuscular blockers—in days; need for a tracheostomy tube; days of respiratory support (i.e., orotracheal intubation or tracheostomy); number of patients in use of an alternative feeding method prior to swallowing assessment; number of patients who received a recommendation to remove alternative feeding method immediately after swallowing assessment; number of rehabilitation sessions to achieve oral feeding and to remove alternative feeding method; time to achieve oral feeding and to remove alternative feeding method (in days); and days to hospital discharge.

## Data analysis

Analysis was performed using SPSS for Windows, version 28.0. In order to show the overall results, categorical variables were presented in contingency tables comprising absolute (n) and relative (%) frequencies, and continuous variables were described using mean and standard deviation. To begin with patients were divided into two groups according to their FOIS level assigned on the last swallowing assessment (i.e., prior to hospital outcome–hospital transfer, hospital discharge or in-hospital death). The groups were characterized as follows: in-hospital resolved dysphagia–patients with FOIS levels 6 and 7; non-resolved dysphagia at hospital outcome–patients with FOIS levels 1 to 5. The groups were compared using the Student T test for continuous data and the Pearson's Chi-square test for categorical data. The adopted significance level was of 5%.

Secondarily, the possible risk factors were analyzed to identify which items were the most significant predictors of non-resolved dysphagia at hospital outcome in the investigated population. The backward stepwise logistic regression model was used to examine the relationships between independent variables. As previously described, the dependent variable was considered non-resolved dysphagia at hospital outcome (i.e., FOIS levels 1 to 5). Any variable having a significant univariate test at $p \leq 0.1$ was selected as a candidate for the multivariate analysis. During the iterative multivariate fitting, covariates were removed from the model if they were non-significant at $p \leq 0.05$ and not a confounder (i.e., did not change any remaining parameter estimates by more than 20%), using the backward stepwise selection method. The variables that remained in the model were considered independent risk factors.

## Results

During the study period, 4,907 adult patients (>18 years) were admitted to the Institution with suspected COVID-19, of whom 1,052 were excluded from this study due to lack of

**Table 2. Demographic and clinical data–intergroup comparison.**

| | Resolved dysphagia (n = 606) | Non-resolved dysphagia (n = 314) | Overall (n = 920) | p-value |
|---|---|---|---|---|
| **Age (years)** (mean±SD) | 53.5 (±14.1) | 62.5 (±13.2) | 56.5 (±14.5) | *<0.001** |
| **Male Sex** n (%) | 338 (55.8%) | 185 (58.9%) | 523 (56.8%) | 0.362 |
| **Type of respiratory support** | | | | |
| Orotracheal intubation n (%) | 550 (90.8%) | 279 (88.9%) | 829 (90.1%) | 0.359 |
| Tracheostomy n (%) | 56 (9.2%) | 35 (11.1%) | 91 (9.9%) | |
| **SAPS-3 score** (mean±SD) | 62.7 (±13.4) | 69.5 (±14.9) | 65.0 (±14.3) | *<0.001** |
| **Body Mass Index** (mean±SD) | 30.1 (±9.0) | 28.9 (±7.1) | 29.6 (±8.2) | *<0.001** |
| **Length of hospital stay (days)** (mean±SD) | 31.7 (±23.4) | 33.2 (±26.9) | 32.2 (±24.7) | 0.380 |
| **PaO2/FiO2 ratio** (mean±SD) | 139.0 (±82.2) | 155.8 (±101.0) | 144.7 (±89.4) | 0.096 |
| **Use of neuromuscular blockers** n (%) | 428 (77.4%) | 228 (77.0%) | 656 (77.3%) | 0.903 |
| **Days of neuromuscular blockers** (mean±SD) | 3.9 (±2.9) | 4.3 (±3.2) | 4.0 (±3.0) | 0.079 |
| **Use of corticosteroids** n (%) | 466 (78.7%) | 249 (80.8%) | 715 (79.4%) | 0.454 |
| **Days of corticosteroids** (mean±SD) | 15.2 (±12.5) | 14.8 (±11.8) | 15.1 (±12.3) | 0.641 |
| **D-Dimer on admission** (mean±SD) | 5830.5 (±11954.4) | 7649.8 (±17453.8) | 6453.1 (±14096.9) | 0.102 |
| **Respiratory support (days)** (mean±SD) | 8.8 (±6.1) | 10.5 (±11.2) | 9.4 (±8.3) | *0.012** |
| **Hospital outcome** | | | | |
| Hospital discharge n (%) | 569 (9.39%) | 154 (49.0%) | 723 (78.6%) | *<0.001*** |
| Hospital transfer n (%) | 19 (3.1%) | 31 (9.9%) | 50 (5.4%) | *<0.001*** |
| In-hospital death n (%) | 18 (3.0%) | 129 (41.1%) | 147 (16.0%) | *<0.001*** |

n: Number of participants; SD: Standard deviation; SAPS-3: *Simplified Acute Physiology Score*–third version; PaO2/FiO2 ratio: Ratio of arterial oxygen partial pressure to fractional inspired oxygen

*significant difference according to Student T test

** significant difference according to Pearson's Chi-square test.

laboratory confirmation of SARS-CoV-2 infection. From the 3,855 eligible patients, 920 (23.9%) were referred for a swallowing assessment. The decision to consult a SLP was left to the discretion of the primary treating physicians. Within this population, 606 patients presented in-hospital resolved dysphagia and 314 presented non-resolved dysphagia at hospital outcome. Demographic and clinical data are presented in Table 2.

According to the intergroup comparison, differences were observed for the following variables: age–patients in the non-resolved dysphagia group were older; severity of illness–patients in the non-resolved dysphagia group presented higher scores on SAPS-3 at admission to the ICU; length of respiratory support–patients in the non-resolved dysphagia group were on respiratory support for more days; and mortality was higher for patients in the non-resolved dysphagia group.

The presence of comorbidities at admission is presented in Table 3. With the exception of vascular diseases and other comorbidities, the groups differed significantly for all the other variables (i.e., patients with non-resolved dysphagia presented a greater number of participants with comorbidities).

Table 4 presents the intergroup comparisons for the bedside swallow evaluation and swallow treatment results (swallowing clinical data). Significant differences were observed for the following variables: FOIS level, both at the initial and outcome assessments and number of patients who removed the alternative feeding method immediately after the swallowing assessment. As expected, patients in the in-hospital resolved dysphagia group presented better results.

Tables 5 and 6 presents the results of the multivariate logistic regression model for the prediction of non-resolved dysphagia at hospital outcome in critical COVID-19 patients. The

**Table 3. Presence of comorbidities at admission–intergroup comparison.**

| | Resolved dysphagia (n = 606) | Non-resolved dysphagia (n = 314) | Overall (n = 920) | p-value |
|---|---|---|---|---|
| **Patients with any comorbidities** n (%) | 510 (84.2%) | 285 (90.8%) | 795 (86.4%) | *0.006** |
| **Description of comorbidities** | | | | |
| Prior acute myocardial infarction n (%) | 18 (3.0%) | 19 (6.1%) | 37 (4.0%) | *0.024** |
| Heart diseases n (%) | 57 (9.4%) | 55 (17.5%) | 112 (12.2%) | *<0.001** |
| Pulmonary diseases n (%) | 62 (10.2%) | 56 (17.8%) | 118 (12.8%) | *0.001** |
| Kidney diseases n (%) | 38 (6.3%) | 45 (14.3%) | 83 (9.0%) | *<0.001** |
| Vascular diseases n (%) | 24 (4.0%) | 16 (5.1%) | 40 (4.3%) | 0.423 |
| Prior stroke n (%) | 13 (2.1%) | 20 (6.4%) | 33 (3.6%) | *0.001** |
| Other neurologic diseases n (%) | 39 (6.4%) | 35 (11.1%) | 74 (8.0%) | *0.013** |
| Diabetes Mellitus n (%) | 195 (32.2%) | 140 (44.6%) | 335 (36.4%) | *<0.001** |
| High blood pressure n (%) | 308 (50.8%) | 201 (64.0%) | 509 (55.3%) | *<0.001** |
| Others n (%) | 344 (56.8%) | 190 (60.5%) | 534 (58.0%) | 0.275 |

n: Number of participants

* significant difference according to Pearson's Chi-square test.

univariate analysis identified 18 covariates initially as potential candidates for the multivariate model at the 0.1 alpha level based on the likelihood-ratio statistic: age, SAPS-3 score, Body Mass Index, PaO2/FiO2 ratio, days of neuromuscular blockers, D-Dimer on admission, length of respiratory support, presence of each comorbidity at admission (except for vascular diseases and other comorbidities), presence of alternative feeding method prior to swallowing assessment, recommendation to remove alternative feeding method after swallowing assessment and FOIS level on initial swallowing assessment.

Table 5 shows the initial results of the logistic regression model, and Table 6 shows the resulting model, containing only significant covariates. This analysis indicated that increasing age, SAPS-3 score, BMI, days of neuromuscular blockers, presence of neurologic diseases (other than stroke), presence of diabetes mellitus, and lower FOIS levels on the initial swallowing assessment were associated with higher chances of presenting dysphagia at hospital outcome. The results point that an increase in age by each year increases the odds of presenting non-resolved dysphagia by 2.7%. Similarly, each point increased in the SAPS-3 score increases the odds of presenting non-resolved dysphagia by 1.9% and an increase in days of

**Table 4. Swallowing data–intergroup comparison.**

| | Resolved dysphagia (n = 606) | Non-resolved dysphagia (n = 314) | Overall (n = 920) | p-value |
|---|---|---|---|---|
| **Number sessions to achieve oral feeding** (mean±SD) | 1.6 (±1.5) | 1.8 (±2.0) | 1.6 (±1.6) | 0.181 |
| **Days to achieve oral feeding** (mean±SD) | 1.5 (±2.8) | 1.9 (±3.5) | 1.6 (±3.0) | 0.117 |
| **Use of AFM prior to assessment** n(%) | 457 (75.4%) | 268 (85.4%) | 725 (78.8%) | *<0.001*** |
| **SLP Recommendation to end AFM after assessment** n(%) | 448 (97.8%) | 105 (38.6%) | 553 (75.8%) | *<0.001*** |
| **Sessions to remove alternative feeding method** (mean±SD) | 3.3 (±3.2) | 3.4 (±4.0) | 3.3 (±3.4) | 0.847 |
| **Days to remove alternative feeding method** (mean±SD) | 5.0 (±6.9) | 5.1 (±6.7) | 5.0 (±6.8) | 0.938 |
| **FOIS level on initial swallowing assessment** (mean±SD) | 4.4 (±1.9) | 2.7 (±1.9) | 3.8 (±2.1) | *<0.001** |
| **FOIS level on hospital outcome** (mean±SD) | 6.7 (±0.5) | 3.5 (±1.9) | 5.7 (±1.9) | *<0.001** |

n: Number of participants; SD: Standard deviation; AFM: Alternative feeding method; SLP: Speech-language pathologist; FOIS: Functional Oral Intake Scale

*significant difference according to Student T test

** significant difference according to Pearson's Chi-square test.

**Table 5. Multivariate logistic regression model for prediction of non-resolved dysphagia at hospital outcome in patients with COVID-19 –first iteration (full model).**

| | Odds Ratio | CI (95%) | | p-value |
|---|---|---|---|---|
| | | Lower | Upper | |
| **Age** | 1.029 | 1.011 | 1.048 | *0.002** |
| **SAPS-3 score** | 1.019 | 1.003 | 1.035 | *0.019** |
| **Body Mass Index** | 0.965 | 0.935 | 0.996 | *0.026** |
| **PaO2/FiO2 ratio** | 1.000 | 0.997 | 1.002 | 0.752 |
| **Days of neuromuscular blockers** | 1.068 | 0.995 | 1.146 | 0.068 |
| **D-Dimer on admission** | 1.000 | 1.000 | 1.000 | 0.646 |
| **Length of respiratory support** | 1.013 | 0.990 | 1.036 | 0.288 |
| **Presence of prior acute myocardial infarction** | 0.728 | 0.285 | 1.861 | 0.507 |
| **Presence of heart diseases** | 0.651 | 0.337 | 1.257 | 0.201 |
| **Presence of pulmonary diseases** | 0.865 | 0.473 | 1.583 | 0.639 |
| **Presence of kidney diseases** | 0.795 | 0.379 | 1.667 | 0.544 |
| **Presence of prior stroke** | 0.594 | 0.200 | 1.763 | 0.348 |
| **Presence of other neurologic diseases** | 1.428 | 1.196 | 1.934 | *0.033** |
| **Presence of Diabetes Mellitus** | 1.608 | 1.381 | 1.971 | *0.037** |
| **Presence of high blood pressure** | 1.398 | 0.860 | 2.272 | 0.176 |
| **Presence of alternative feeding method prior to swallowing assessment** | 0.954 | 0.523 | 1.739 | 0.877 |
| **Recommendation to remove alternative feeding method after swallowing assessment** | 1.187 | 0.191 | 7.384 | 0.854 |
| **FOIS level on initial swallowing assessment** | 0.734 | 0.658 | 0.819 | *<0.001** |

CI: Confidence interval; FOIS: Functional Oral Intake Scale; SAPS-3: *Simplified Acute Physiology Score*–third version; PaO2/FiO2 ratio: Ratio of arterial oxygen partial pressure to fractional inspired oxygen

*significant interaction according to multivariate logistic regression–full model.

neuromuscular blockers by each day increases the odds of presenting non-resolved dysphagia by 7.6%. Increases in the BMI reduced the odds of presenting non-resolved dysphagia by 4%. Patients with neurologic diseases and with diabetes mellitus are 1.44 and 1.65 and times more likely to present non-resolved dysphagia at hospital outcome, respectively. Lower FOIS level on the initial swallowing assessment (by each level decrease) increase the odds of presenting non-resolved dysphagia at hospital outcome by 72,5%.

**Table 6. Multivariate logistic regression model for prediction of non-resolved dysphagia at hospital outcome in patients with COVID-19 – 12th and last iteration (resulting model).**

| | Odds Ratio | CI (95%) | | p-value |
|---|---|---|---|---|
| | | Lower | Upper | |
| **Age** | 1.027 | 1.010 | 1.044 | *0.002** |
| **SAPS-3 score** | 1.019 | 1.004 | 1.035 | *0.015** |
| **Body Mass Index** | 0.960 | 0.931 | 0.990 | *0.008** |
| **Days of neuromuscular blockers** | 1.076 | 1.008 | 1.148 | *0.028** |
| **Presence of other neurologic diseases** | 1.443 | 1.206 | 1.956 | *0.038** |
| **Presence of Diabetes Mellitus** | 1.646 | 1.423 | 1.986 | *0.043** |
| **FOIS level on initial swallowing assessment** | 0.725 | 0.654 | 0.803 | *<0.001** |

CI: Confidence interval; FOIS: Functional Oral Intake Scale; SAPS-3: *Simplified Acute Physiology Score*–third version

*significant interaction according to multivariate logistic regression–backward stepwise selection method.

## Discussion

Overall, our results indicated that patients included in the study had a mean age of 56.5 years, were generally overweight and had a high prevalence of comorbidities. The majority of these patients (65.9%), despite the use of prolonged orotracheal intubation and/or use of a tracheostomy tube, were discharged from hospital with few to no oral intake restrictions of food and liquid. A few variables were independently associated with non-resolved dysphagia at hospital outcome: increasing age, higher SAPS-3 score at admission, BMI, days of neuromuscular blockers, presence of neurologic diseases (other than stroke), presence of diabetes mellitus, and lower FOIS levels on the initial swallowing assessment.

In our study, age was a predictive factor for the non-resolution of dysphagia in COVID-19 patients. Changes in swallowing have been extensively studied in the elderly [30]. According to the literature, the aging process can have a negative impact on the swallowing function, and dysphagia is pointed as a frequent condition in the elderly population [30,31]. Older individuals generally demonstrate a high prevalence of dysphagia because of underlying diseases and age-related changes. It is estimated that up to 20% of individuals over the age of 50 and most individuals over the age of 80 have some degree of dysphagia [31]. Changes in swallowing physiology, such as loss of muscle mass and elastic connective tissue properties, can result in loss of muscle strength and mobility [32]. These changes can have a negative impact on the swallowing efficiency and airway protection. Age-related atrophy of the soft tissues of the pharynx and larynx may also be considered as a contributing factor to swallowing changes [31]. Moreover, causes of dysphagia in the elderly have been attributed to the presence of neurological diseases (e.g., stroke and dementia) and neuromuscular disorders (e.g., hypo and hyperthyroidism and peripheral neuropathy secondary to diabetes), among others [31]. Considering COVID-19, studies have already described that increasing age is a strong predictive factor for a stronger and faster loss of muscle activities, increasing the incidence of dysphagia and mortality [33].

The identification of high-risk patients admitted to intensive care units with COVID-19 can direct treatment strategies and reduce clinical complications related to dysphagia. In our study, the severity of illness on admission to the ICU was an important factor that differentiated patients with resolved and non-resolved dysphagia. Patients with high SAPS-3 scores on admission presented poorer swallowing outcomes. Other studies have already pointed the correlation between the severity of illness and dysphagia in patients submitted to prolonged orotracheal intubation [15]. For patients with COVID-19, high SAPS-3 scores have been associated with the need for advanced respiratory support and intensive care [34], which characterizes risk factors for the development of dysphagia [15].

Most of the population in the present study had a BMI indicating overweight or obesity (average of 29.6 kg/m$^2$). Patients in the resolved dysphagia group presented a mean BMI indicative of obesity (30.1 kg/m$^2$), while patients in the non-resolved dysphagia group presented a mean BMI indicative of overweight (28.9 kg/m$^2$). A recent study of septic patients in the ICU demonstrated a lower risk of mortality for obese individuals when compared to normal, overweight or underweight patients [35]. In our study, patients in the resolved dysphagia group presented more favorable outcomes. Although most studies with COVID-19 patients indicated that obesity was prevalent in hospitalized patients and that a higher BMI was associated with higher in-hospital mortality, studies are contradictory regarding the association of BMI with the severity of critical illness secondary to COVID-19 [33,36]. In our institution, the mortality rate in patients with COVID-19 who were submitted to mechanical ventilation was of 44%. However, BMI was not associated to increased mortality [37].

Regarding the presence of comorbidities, pre-existing conditions affecting general health are described in the literature as potential risk factors that predict the course and severity of

COVID-19, increasing the risk of clinical complications, orotracheal intubation and death [38,39]. A meta-analysis pointed out that hypertension, diabetes, chronic obstructive pulmonary disease (COPD), cardiovascular disease, and cerebrovascular disease are the main risk factors for complications in patients with COVID-19 [40]. In the present study, the presence of Diabetes Mellitus, hypertension, acute myocardial infarction, heart disease, lung disease, kidney disease, stroke, other neurological diseases and higher death rates were associated with the group of non-resolved dysphagia. Our result corroborates those of other studies that indicated the prevalence of dysphagia in various pathologies such as heart disease, pulmonary disease, stroke and other neurologic diseases [41,42].

It is important to highlight that the presence of Diabetes Mellitus was independently associated with non-resolved dysphagia. Although this subject is still unexplored, a few studies have reported that swallowing complaints are common findings in patients with diabetes, especially in patients who present autonomic and/or peripheral neuropathy, a common complication of this disease [43]. The prevalence of Diabetes Mellitus in the present study was of 36.4%, a rate similar do the one found in the literature [44]. Moreover, studies have already pointed that diabetes in patients with COVID-19 is associated with a twofold increase in mortality and severity when compared to non-diabetic patients [45]. The presence of diabetes usually worsens the prognosis of infection and is related to a higher morbidity and mortality from sepsis when compared to the general population [46]. Given the relevance of such finding, it is suggested that further studies be conducted in order to better explore the relationship between Diabetes Mellitus and dysphagia in COVID-19 patients and also in other populations.

Considering the resolution of dysphagia, previous research have pointed a significant correlation between the scores on the functional scales of swallowing at the beginning of rehabilitation and at hospital discharge [14,15,47]. Similar results were found in our study, indicating that patients with better swallowing scores on the initial swallowing assessment were discharged without any swallowing difficulties, whereas patients with a greater compromise of swallowing on the initial assessment were discharged from hospital with significant eating restrictions and were referred to swallowing rehabilitation services.

A large proportion of the severely affected patients with COVID-19 used medications such as neuromuscular blockers and corticosteroids due to severe respiratory alterations, need for mechanical ventilation, prone position and the inflammatory response to the disease [48]. The use of these drugs is known to be significantly associated with ICU acquired muscle weakness and neuropathies, and therefore can increase the risk of developing dysphagia [49]. Swallowing is a sequence of complex sensory-motor activities and any interruption in this mechanism can cause alterations in the swallowing process. Although studies have indicated that muscle strength of patients with COVID-19 improves during hospitalization, the impact on functional status tends to remain substantial, emphasizing the need of providing personalized rehabilitation to reduce the long-term impacts of the disease. Based on these findings, we hypothesized that the use of corticosteroids and neuromuscular blockers would have an impact on the swallowing of our patients. However, the results only indicated the association between the use of neuromuscular blockers and the resolution of dysphagia during hospitalization.

Although the sample of participants included in the present study was derived from a single institution and the results may reflect the characteristics of the procedures adopted at this location, it is the study with the largest sample of participants to investigate the swallowing outcomes in critical COVID-19 patients. ICU performance has never been more important or more difficult than during the COVID-19 pandemic. The COVID-19 pandemic required fast adaptations in ICU and patient management. This fact, associated with the lack of effective and specific treatment protocols, lack of adequate equipment to provide life support, under-staffing, late patient referral most definitely had an impact on the outcome of patients. Further

studies should continue to be conducted to better elucidate the clinical and prognostic findings, as well as the rehabilitation outcomes in critical patients affected by COVID-19.

## Acknowledgments

The authors would also like to acknowledge the contribution of Julia do Prado Amarilla Rojas during data gathering, as part of her training as junior researcher.

## Author Contributions

**Conceptualization:** Claudia Regina Furquim de Andrade.

**Data curation:** Fernanda Chiarion Sassi, Ana Paula Ritto, Maíra Santilli de Lima, Cirley Novais Valente Junior, Paulo Francisco Guerreiro Cardoso, Bruno Zilberstein, Paulo Hilário Nascimento Saldiva.

**Formal analysis:** Ana Paula Ritto.

**Investigation:** Maíra Santilli de Lima, Cirley Novais Valente Junior.

**Methodology:** Fernanda Chiarion Sassi, Claudia Regina Furquim de Andrade.

**Project administration:** Fernanda Chiarion Sassi, Claudia Regina Furquim de Andrade.

**Writing – original draft:** Fernanda Chiarion Sassi, Ana Paula Ritto, Maíra Santilli de Lima, Cirley Novais Valente Junior, Claudia Regina Furquim de Andrade.

**Writing – review & editing:** Fernanda Chiarion Sassi, Ana Paula Ritto, Paulo Francisco Guerreiro Cardoso, Bruno Zilberstein, Paulo Hilário Nascimento Saldiva, Claudia Regina Furquim de Andrade.

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
