## [Decision Letter · Decision Letter 0]

21 Apr 2022

PONE-D-22-04075Characteristics of postintubation dysphagia in ICU patients in the context of the COVID-19 outbreak: A report of 920 cases from a Brazilian reference centerPLOS ONE

Dear Dr. de Andrade,

Thank you for submitting your manuscript to PLOS ONE. After careful consideration, we feel that it has merit but does not fully meet PLOS ONE’s publication criteria as it currently stands. Therefore, we invite you to submit a revised version of the manuscript that addresses the points raised during the review process.

We look forward to receiving your revised manuscript.

Kind regards,

Chiara Lazzeri

Academic Editor

PLOS ONE

Journal Requirements:

Upon re-submitting your revised manuscript, please upload your study’s minimal underlying data set as either Supporting Information files or to a stable, public repository and include the relevant URLs, DOIs, or accession numbers within your revised cover letter. For a list of acceptable repositories, please see http://journals.plos.org/plosone/s/data-availability#loc-recommended-repositories. Any potentially identifying patient information must be fully anonymized

Reviewers' comments:

Reviewer's Responses to Questions

**Comments to the Author**

1. Is the manuscript technically sound, and do the data support the conclusions?

Reviewer #1: Partly

Reviewer #2: Yes

2. Has the statistical analysis been performed appropriately and rigorously? 

Reviewer #1: I Don't Know

Reviewer #2: I Don't Know

3. Have the authors made all data underlying the findings in their manuscript fully available?

Reviewer #1: Yes

Reviewer #2: Yes

4. Is the manuscript presented in an intelligible fashion and written in standard English?

Reviewer #1: Yes

Reviewer #2: Yes

5. Review Comments to the Author

Reviewer #1: Authors realized, and proposed for publication, a monocentric prospective observational cohort study to determine risk factors of swallowing dysfunction in COVID 19 patients who required prolonged mechanical ventilation. We thank them for this work.

General remarks:

Authors realized a significant work in view of the number of patients included, but restricting ourselves to patients with COVID 19 raises the question of the external validity of the results. This work also confirms the possibility of relying on a method of clinical evaluation of swallowing.

Detailed comments:

Introduction :

P5, l 77-78: another recent reference can be reported (https://doi.org/10.1186/s13613-020-00752-x); this also opening the discussion to be carried out on the definition of an extended duration of mechanical ventilation.

P6, l 86-88 : lacks a reference comparing COVID 19 and non-COVID 19 patients; or specify that such a reference does not exist.

Methods :

Specify how many beds are available in the dedicated unit.

P7 l 108 : why this extremely questionable choice of 48 hours to define a prolonged orotracheal intubation? Showing the results with other limits seems essential.

Results:

On what criteria are the only 20% of patients eligible for this swallowing evaluation chosen? This is a crucial point.

Many results, of unequal interest. Simplify the presentation. Those of table 3 could be proposed in appendix.

Table 4: it is unclear to what clinical situation these results correspond

Discussion :

To be reviewed once the presentation of the results has been revised and simplified. The discussion will then have to be shortened.

So authors propose us an interesting monocentric prospective observational study, which need important modification before hoping for publication.

Reviewer #2: Many thanks for the opportunity to act as a reviewer of this work. Congratulations, the work is well written and covers the important part of our practice these days. Several comments:

-- I am missing more structured description of the swallowing management.

-- Why do you think there is such a large significant difference between the groups for the presence of commorbidities such as kidney diseases or high blood pressure. Please explain in the discussion.

-- The presence of tracheostomy plays a crucial role in dysphagia. Would you consider it to be factor for bad prognostic outcome in your study?

Many thanks.

6. PLOS authors have the option to publish the peer review history of their article (what does this mean?). If published, this will include your full peer review and any attached files.

Reviewer #1: No

Reviewer #2: No

---

## [Author Response · Author response to Decision Letter 0]

18 May 2022

São Paulo, 2nd May 2022 

Prof. Emily Chenette

Editor-in-Chief

PLOS ONE

Ref.: Manuscript ID PONE-D-22-04075

We thank PLOS ONE for the careful and sensible peer review of our manuscript entitled “Characteristics of postintubation dysphagia in ICU patients in the context of the COVID-19 outbreak: A report of 920 cases from a Brazilian reference center”. All of the suggestions in the review enabled us to elucidate issues that were not as clear as we thought they were. 

Based on the suggestions made by reviewers #1 and #2, we are now submitting the revised version for your appreciation. As requested, all modifications made in the original text are marked by a red colored text as follows:

Reviewer #1:

General remarks:

Authors realized a significant work in view of the number of patients included, but restricting ourselves to patients with COVID 19 raises the question of the external validity of the results. This work also confirms the possibility of relying on a method of clinical evaluation of swallowing.

The purpose of this study, as mentioned in the manuscript, was “to identify risk factors that were independently related to the maintenance of a swallowing dysfunction in patients affected by critical COVID-19 and who required prolonged mechanical ventilation after discharge from the ICU”. Therefore, we do not understand how including only patients with COVID-19 would be restricting the external validity of the results. We agree that in its current format, it would only be applicable to COVID-19 patients, but we do not see that as a weakness as it was the purpose of the study.

Detailed comments:

Introduction:

P5, l 77-78: another recent reference can be reported (https://doi.org/10.1186/s13613-020-00752-x); this also opening the discussion to be carried out on the definition of an extended duration of mechanical ventilation.

We updated the manuscript to include this suggestion. 

P6, l 86-88: lacks a reference comparing COVID 19 and non-COVID 19 patients; or specify that such a reference does not exist.

We updated the manuscript to include this suggestion.

Methods:

Specify how many beds are available in the dedicated unit.

We updated the manuscript to include this suggestion.

P7 l 108: why this extremely questionable choice of 48 hours to define a prolonged orotracheal intubation? Showing the results with other limits seems essential.

We updated the manuscript to make this information clearer to the reader. 

An extensive set of studies define prolonged tracheostomy as intubation for 48 hours or longer. We included eight more references to support the choice of 48 hours to define prolonged orotracheal intubation. 

Results:

On what criteria are the only 20% of patients eligible for this swallowing evaluation chosen? This is a crucial point.

The decision to consult a SLP was left to the discretion of the primary treating physicians. We updated the manuscript to include this suggestion.

Many results, of unequal interest. Simplify the presentation. Those of table 3 could be proposed in appendix.

We updated the manuscript to include this suggestion partially, but we believe the data presented in table 3 is relevant for the discussion, and therefore choose to maintain it in the main manuscript and not as an appendix. The presence of many of the comorbidities were significantly different between groups, and we believe that is an interesting and relevant finding.

Table 4: it is unclear to what clinical situation these results correspond

Table 4 presents the specific information regarding the bedside swallow evaluation and swallow treatment performed by the speech-language pathologist. We updated the manuscript to include this suggestion.

Discussion :

To be reviewed once the presentation of the results has been revised and simplified. The discussion will then have to be shortened.

We did update the manuscript to simplify the presentation of the results, as suggested. Nevertheless, the results remain the same, just presented differently, so we do not understand why the discussion should be changed. Only the significant findings were included in our discussion. 

 

Reviewer #2:

Several comments:

-- I am missing more structured description of the swallowing management.

We updated the manuscript to include this suggestion.

-- Why do you think there is such a large significant difference between the groups for the presence of commorbidities such as kidney diseases or high blood pressure. Please explain in the discussion.

We updated the manuscript to include this suggestion.

-- The presence of tracheostomy plays a crucial role in dysphagia. Would you consider it to be factor for bad prognostic outcome in your study?

In our study, tracheostomy was not a significant factor for the non-resolved dysphagia at hospital outcome, as there was no significant difference between outcome groups for the presence of tracheostomy, as presented in Table 1.

Thank you for the opportunity of revising our paper.

Regards,

Prof. Claudia Regina Furquim de Andrade (on behalf of all authors)

---

## [Editor Report · Decision Letter 1]

6 Jun 2022

Characteristics of postintubation dysphagia in ICU patients in the context of the COVID-19 outbreak: A report of 920 cases from a Brazilian reference center

PONE-D-22-04075R1

Dear Dr. de Andrade,

We’re pleased to inform you that your manuscript has been judged scientifically suitable for publication and will be formally accepted for publication once it meets all outstanding technical requirements.

Kind regards,

Chiara Lazzeri

Academic Editor

PLOS ONE
---

## [Editor Report · Acceptance letter]

9 Jun 2022

PONE-D-22-04075R1 

Characteristics of postintubation dysphagia in ICU patients in the context of the COVID-19 outbreak: A report of 920 cases from a Brazilian reference center 

Dear Dr. de Andrade:

I'm pleased to inform you that your manuscript has been deemed suitable for publication in PLOS ONE. Congratulations! Your manuscript is now with our production department. 

Kind regards, 

on behalf of

Dr. Chiara Lazzeri 

Academic Editor

PLOS ONE